# Evaluating General Purpose Vision Foundation Models for Medical Image Analysis: An Experimental Study of DINOv2 on Radiology Benchmarks

## Abstract

The integration of deep learning systems into healthcare has been hindered by the resource-intensive process of data annotation and the inability of these systems to generalize to different data distributions. Foundation models, which are models pre-trained on large datasets, have emerged as a solution to reduce reliance on annotated data and enhance model generalizability and robustness. DINOv2 is an open-source foundation model pre-trained with self-supervised learning on 142 million curated natural images that exhibits promising capabilities across various vision tasks. Nevertheless, a critical question remains unanswered regarding DINOv2's adaptability to radiological imaging, and whether its features are sufficiently general to benefit radiology image analysis. Therefore, this study comprehensively evaluates the performance DINOv2 for radiology, conducting over 200 evaluations across diverse modalities (X-ray, CT, and MRI). To measure the effectiveness and generalizability of DINOv2's feature representations, we analyze the model across medical image analysis tasks including disease classification and organ segmentation on both 2D and 3D images, and under different settings like kNN, few-shot learning, linear-probing, end-to-end fine-tuning, and parameter-efficient fine-tuning. Comparative analyses with established supervised, self-supervised, and weakly-supervised models reveal DINOv2's superior performance and cross-task generalizability. The findings contribute insights to potential avenues for optimizing pre-training strategies for medical imaging and enhancing the broader understanding of DINOv2's role in bridging the gap between natural and radiological image analysis.

## 1 Introduction

Radiology imaging plays a pivotal role in modern medicine, serving as an indispensable tool for accurate and timely diagnosis (Hussain et al., 2022). The importance of radiological imaging lies in its ability to provide detailed and non-invasive visualizations (Peng et al., 2022) of the internal structures and functions of the human body. Deep learning-based computer vision methods have been successful at analyzing and processing radiological imaging, leading to systems that can extract clinically-relevant information with high accuracy (Rajpurkar et al., 2017). However, the success of these systems has been reliant on annotated medical data, which is expensive to obtain because it requires the time and effort of trained radiologists (Khan et al., 2023). Moreover, these systems can achieve high-level accuracy only within a specified scope, but fail to generalize across domains, tasks, and slight data distribution shifts (Kelly et al., 2019).

Recently, the field of computer vision has seen a rise in interest for general-purpose models that are optimized to function across different tasks and domains (Yuan et al., 2021; Radford et al., 2021; Oquab et al., 2023; Kirillov et al., 2023). These models, grouped under the term "Foundation Models" (FMs), usually contain parameters ranging from hundreds of millions to tens of billions and are trained on large datasets, on the order of tens of millions. As a result of this large-scale training, these FMs often achieve state-of-the-art (SoTA) results and impressive zero-shot and few-shot performance and generalizability (Oquab et al., 2023; Kirillov et al., 2023). For these reasons, foundation models have gained traction in deep learning-based

Relative Performance for Classification and Segmentation

Figure 1: Cross-task generalizability of DINOv2 compared to other models. The horizontal axis shows the relative performance or ranking of each of the models on segmentation tasks, while the vertical axis does the same for classification tasks. The ranking is calculated as the average ranking across all segmentation (5 Datasets) or classification tasks (4 Datasets), where rank 1 means the model performs the best relative to other models. Models pre-trained with weakly-supervised learning perform well only on classification tasks, while MAE performs well only on segmentation. DINOv2 can generalize across both tasks and outperforms all other models for classification.

medical image analysis research (Zhou et al., 2023; Wang et al., 2023; Caro et al., 2023; Qiu et al., 2023; Ma et al., 2023), as they hold promise for reducing the reliance on the expensive process of annotating medical data and towards the goal of building generalist medical artificial intelligence systems that can function across a variety of tasks and domains (Moor et al., 2023).

## 1.1 What Are Foundation Models?

The term "Foundation Model" encompasses a broad spectrum of models that may initially appear distinct. In the most general sense, foundation models are large models trained on large datasets and can generalize across tasks and/or domains (Merritt, 2023). To make the term more useful in our analysis, we categorize foundation models using two distinct methods. First, we divide FMs depending on their training paradigm into three groups: self-supervised, weakly-supervised, and supervised foundation models. Weakly-supervised and supervised foundation models require correspondence in the training data. In these paradigms, the training data is required to be available in pairs: an X-ray examination and a corresponding interpretation, diagnosis, or segmentation mask for example. Models like OpenAI's CLIP (Radford et al., 2021) and Meta's Segment Anything Model (SAM) (Kirillov et al., 2023) fall under this category. Self-supervised foundation models, on the other hand, require one input data to train (an image, text, audio, etc.) Meta's DINOv2 (Oquab et al., 2023) and Google's Universal Speech Model (USM) (Zhang et al., 2023) belong to this category.

Additionally, we categorize FMs into two groups depending on the generalizability of their produced representations: general purpose (also called task-agnostic), and task-specific FMs. General purpose foundation models produce features that generalize across more than one task, segmentation and classification for example, while task-specific models specialize on only one task. DINOv2 (Oquab et al., 2023) and USM (Zhang et al., 2023) fall under the former category, while SAM (Kirillov et al., 2023) is under the latter.

Because of the less restrictive training dependencies for self-supervised FMs, we believe that they are a promising option to explore for future research especially for the medical imaging domain, since the shortage of training data is one of the main constraints for applications in the field (Khan et al., 2023). Reducing the requirement for annotated data significantly increases the set of possible data used for training, and using unannotated data makes it possible to combine data that are paired with different labels (for example, combining those with class labels and those with segmentation masks). Moreover, FMs that produce representations that can be used across a variety of tasks are desired because it is usually a signal of model robustness. As a result, we focus our attention on self-supervised general-purpose FMs for medical image analysis. Specifically, we adopt Meta's DINOv2 (Oquab et al., 2023) model, a publicly available general-purpose vision foundation model that can extract robust representations across different vision tasks, for experimentation on a wide range of medical disease classification and organ segmentation benchmarks, across different radiological exams and under different evaluation settings.

## 1.2 What is DINOv2?

DINOv2 is a successor of DINO (Caron et al., 2021) and constitutes both a self-supervised pre-training method based on DINO and iBOT (Zhou et al., 2022), and a collection of models pre-trained using that method. It was released to the public by Meta in April 2023 and promises robust representations that enable general-purpose functionality with visual-only data (Oquab et al., 2023). The released models were pre-trained on a dataset of 142 million carefully curated natural images, called LVD-142M. Roughly 100 million of these are images that are similar to ImageNet, curated from calculating similarly of web-scarped images with ImageNet21k dataset (Deng et al., 2009). The remaining images were retrieved based on their similarity to Caltech 101 (Fei-Fei et al., 2004), ADE20k (Zhou et al., 2017), and Google Landmarks v2 (Weyand et al., 2020), among others. Since the LVD-142M dataset was built by scraping images that are similar to natural image datasets (ImageNet21K, ADE20K, etc.), there was likely very little to no medical images scraped into in the dataset, since medical images have distributions that are different from natural images. Hence, DINOv2 likely did not see any of the images in the datasets evaluated.

The models achieve competitive performance on classification, segmentation, depth estimation, and image retrieval tasks across both image and video benchmarks. Moreover, because of the adopted discriminative self-distillation approach, DINOv2 performs well "out-of-the-box" without the need to fine-tune the encoder. The giant version of the model (ViT-g/14) achieves an accuracy of 86.5% and 83.5% on Linear-probing and kNN evaluations, respectively, on ImageNet-1k, outperforming other weakly-supervised and self-supervised methods. This capability to perform well out-of-the-box is appealing, especially in the medical domain, as it implies competitive performance even in low-data and low-computation settings.

## 1.3 Contribution

In this paper, we set to work towards vision-only general-purpose foundation models for the medical domain by adopting DINOv2 for disease classification and organ segmentation benchmarks. We perform comprehensive evaluations of DINOv2 across various scenarios for multiple radiology modalities, exploring both low (few-shot) and high data settings spanning X-ray, CT, and MRI examinations. We benchmark DINOv2's performance with other natural-image supervised, self-supervised, and weakly-supervised models like CLIP, OpenCLIP, and SAM, among others shown in Table 2.

We evaluate DINOv2 on both disease classification and organ segmentation benchmarks. For disease classification tasks, we evaluate the model on kNN, linear-probing, few-shot learning, parameter-efficient fine-tuning, and end-to-end fine-tuning scenarios. For organ segmentation tasks, we compare lightweight and non-lightweight decoders, while we keep the DINOv2 backbone frozen. We compared the results to supervised, weakly-supervised, and self-supervised models. As far as we know, there is no comprehensive

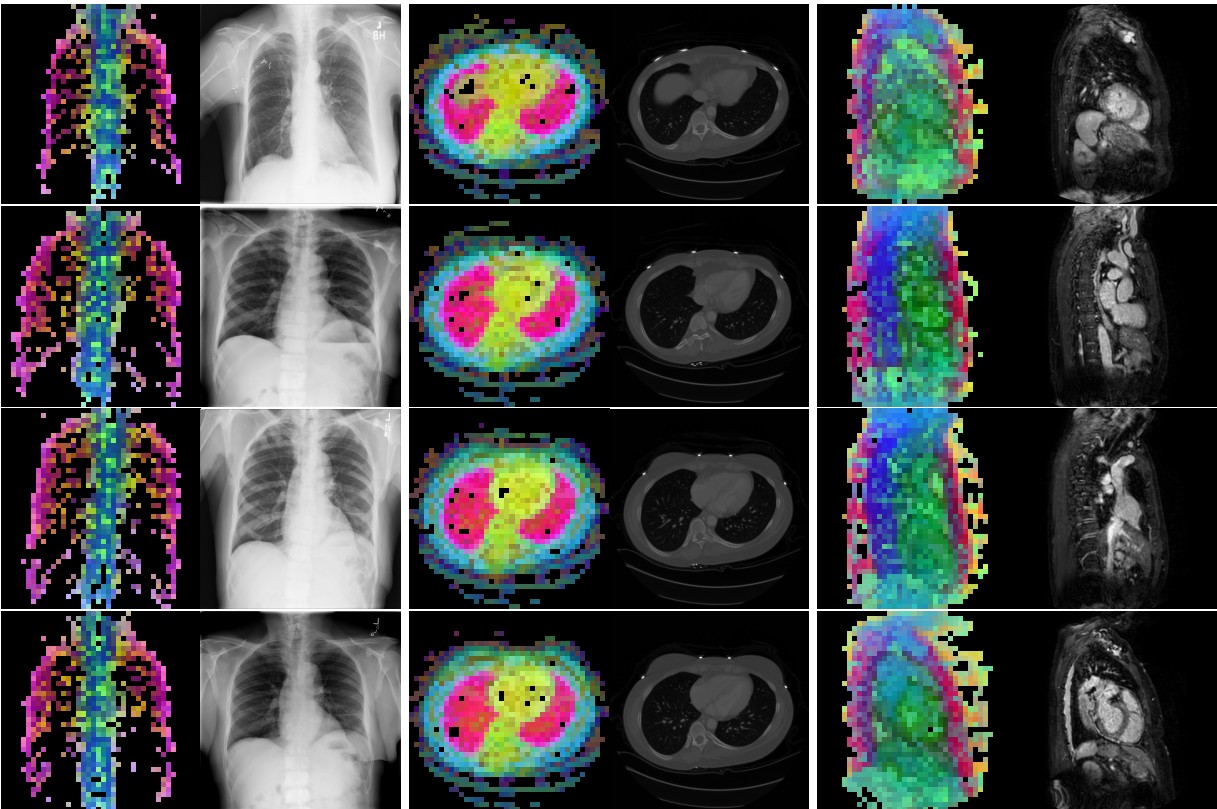

Figure 2: **PCA component visualization.** Following Oquab et al. (2023), the PCA is computed between patches of scans that are in the same column, and the first 3 components are shown. Thresholding is used on the first component to remove the background. Just like in natural images [8], the colors of the three PCA components correspond well with the same parts of images in the same category. This is an easier task however, compared to natural images, because there is less variability between examinations on medical images compared to natural images.

analysis of DINOv2 on medical benchmarks that evaluates the model across disease classification and organ segmentation tasks and on different radiological examinations. Our contributions can be summarized as follows:

- Evaluate DINOv2 on disease classification and organ segmentation radiology benchmarks, under different evaluation settings. We conclude that DINOv2 outperforms self-supervised and weakly-supervised models on classification tasks and is competitive with supervised models pre-trained on ImageNet21K. For segmentation tasks, we observe that DINOv2 outperforms weakly-supervised methods by a large margin, and outperforms or is competitive with supervised methods trained end-to-end.

- Analyze the cross-task generalizability of DINOv2 compared to supervised, self-supervised, and weakly-supervised models. Our analysis concludes that weakly-supervised models perform well on classification tasks only, and masked imaged modeling self-supervision performs well only on segmentation tasks, while DINOv2 can generalize across both tasks. DINOv2 also outperforms supervised classification models when evaluated on segmentation tasks, and outperforms SAM for disease classification.

- Employ parameter-efficient fine-tuning with a DINOv2 ViT-g/14 model on large X-ray classification datasets and provide a performance and efficiency comparison of PEFT with end-to-end fine-tuning

and linear-probing. We conclude that using PEFT can yield performance that is competitive with end-to-end fine-tuning using less than 1% of the total model parameters.

## 2 Related Works

**Supervised foundation models.** Training a large-scale network with supervised learning is non-trivial because it is difficult to combine different labels together. This is even more true in the medical domain, where labeled data is scarce. Even still when labels are seemingly similar, other issues can arise. Cohen et al. (2020) have attempted to train a large-scale X-ray foundation model with supervised learning by combining public chest X-ray disease classification datasets that share diseases. their analysis highlights the fact that discrepancies in labeling caused by disease concept shift across hospitals and observers can lead to worse generalizations for deep learning models. Others have also tried supervised approaches, but the data annotation limitation is still a factor (Mustafa et al., 2021).

Regardless, there successful attempts at building a large-scale supervised foundation model. The Segment Anything Model (SAM) (Kirillov et al., 2023), a supervised foundation model for prompt-based segmentation trained with automatically extracted segmentation masks, is one of those attempts. Since the release of SAM, its applicability for the medical images has been explored (He et al., 2023; Zhang & Jiao, 2023; Mazurowski et al., 2023). Its performance for medical segmentation was found to be inadequate in most cases and highly variable. Because of that, Ma et al. (2023) have developed MedSAM by adopting SAM for the medical domain using a dataset of more than one million medical images across diverse segmentation tasks and examination modalities. MedSAM significantly outperforms SAM on medical tasks and achieves comparable performance to U-Net (Ronneberger et al., 2015) models. Still, MedSAM is limited to medical segmentation, and it is unclear whether it can be generalized to other medical image analysis tasks.

**Weakly-supervised foundation models.** Previous research has explored weakly-supervised vision-language models for zero-shot classification, visual question answering, and image generation in the medical domain, achieving SoTA or competitive performance on radiology benchmarks (Wang et al., 2022; Chambon et al., 2022b; Zhang et al., 2022; Chambon et al., 2022a). Zhang et al. (2022) applied contrastive learning to paired X-ray image-report for learning visual representations. However, the availability of corresponding image-text pairs is a stringent requirement to be applicable for most medical datasets. Moreover, as highlighted by Wang et al. (2022), a false negative issue exists when applying this learning paradigm in the medical domain because, unlike diverse natural images, medical reports can describe other patients that are not necessarily paired. Instead, Wang et al. (2022) proposed MedCLIP, employing an extension of CLIP's contrastive pre-training method that utilizes unpaired examination-report samples, using the MIMIC-CXR (Johnson et al., 2019) and CheXpert (Irvin et al., 2019) datasets. They outperform previous SoTA on zero-shot and supervised classification on four radiology benchmarks. Yet, the performance of MedCLIP, along with other vision-language models, is still limited by the availability of text-image data.

**Self-supervised foundation models.** Azizi et al. (2021) has shown that SSL pre-training on natural datasets can improve performance on medical datasets, even with the domain gap. More recently, self-supervised pre-training applied to medical datasets has recently achieved SoTA on CT and MRI segmentation benchmarks (Valanarasu et al., 2023; Tang et al., 2022). For example, Tang et al. (2022) pre-trained a Swin UNETR (Hatamizadeh et al., 2022) architecture with self-supervision on 5,050 CT volumes and outperformed previous SoTA on the BTCV (Landman et al., 2015) and MSD (Antonelli et al., 2022) competitions. However, to the best of our knowledge, there is still no self-supervised general-purpose FM for the medical domain that has achieved consistently competitive results across tasks and radiological examinations.

## 3 Methodology

In this section, we describe the motivation for adopting DINOv2 for this study and outline the settings under which we performed our experiments. All the details about the preprocessing and hyperparameter tuning pipeline are publicly available in our code (available in Section 7).

## 3.1 Motivation for Using DINOv2

There are many vision foundation models trained with supervised, self-supervised, and weakly-supervised learning on natural images (Radford et al., 2021; Kirillov et al., 2023; Yuan et al., 2021; Xiao et al., 2023; Ilharco et al., 2021). However, we decided to employ DINOv2 in our analysis because of the robustness of its representations, achieving competitive performance in multiple downstream tasks, across vision modalities (image and video), and, most importantly, in out-of-the-box evaluations Oquab et al. (2023). One reason for such performance comes from the fact that the DINOv2 method optimizes for both high-level and low-level features simultaneously, with the incorporation of a patch-level objective to the original DINO Caron et al. (2021). The DINOv2 training paradigm was also specifically designed to generate powerful representations on out-of-the-box kNN evaluations, and outperforms many other weakly-supervised and self-supervised foundation models in kNN and linear-probing (Oquab et al., 2023).

## 3.2 Datasets

We evaluated DINOv2 on 9 public radiology benchmarks, spanning X-ray, CT, and MRI examinations (Exam.) for disease classification (CLS, 4 datasets) and organ segmentation (SEG, 5 datasets) tasks. A summary of the used datasets is shown in Table 1. "# Classes" describes the number of classes and "# Images/Volumes" describes the number of images for 2D datasets and volumes for 3D datasets, respectively. They describe the number of images or volumes that we use for each dataset, and not the number in the original dataset. This is because some of the datasets used have subsets that are not publicly accessible, like the test sets in MSD (Antonelli et al., 2022). Moreover, we only used a subset of AMOS (Ji et al., 2022) because of computational constraints, and only selected the frontal views from CheXpert (Irvin et al., 2019).

Moreover, some of the datasets used do not have a predefined test set. On these datasets, we used systemic sampling to divide the dataset into train, evaluation, and test subsets. All the datasets used are publicly available, and we provide a link for downloading our specific train, validation, and test split in Section 7. Moreover, The data preprocessing pipeline is also available in the GitHub repository, and we describe it briefly in Appendix A.2.

The datasets were chosen based on the diversity of tasks and modalities, and not on their clinical usefulness or value. Classifying different brain tumors types might not be a clinically applicable task, but we still included it in our analysis to gauge how each model performs in disease classification with MRI examinations.

Table 1: The datasets used. For datasets that do not have a standardized test set, we chose the test set using systematic sampling. All the datasets and splits are public.

| Dataset | Exam. | Task | Labels | # Classes | # Images/Volumes | Dim |
|---|---|---|---|---|---|---|
| NIH Chest X-ray (Wang et al., 2017) | X-ray | CLS | Thorax Diseases | 14 | 112,120 | 2D |
| CheXpert (Irvin et al., 2019) | X-ray | CLS | Thorax Diseases | 5 | 161,792 | 2D |
| Montgomery County (MC) (Jaeger et al., 2014) | X-ray | SEG | Lung | 3 | 138 | 2D |
| SARS-CoV-2 (Soares et al., 2020) | CT | CLS | COVID-19 Diagnosis | 2 | 210 | 3D |
| AMOS (Ji et al., 2022) | CT | SEG | Abdominal Organs | 15 | 150 | 3D |
| MSD Spleen (Antonelli et al., 2022) | CT | SEG | Spleen | 2 | 40 | 3D |
| MSD Hipp (Antonelli et al., 2022) | MRI | SEG | Hippocampus Head and Body | 3 | 260 | 3D |
| MSD Heart (Antonelli et al., 2022) | MRI | SEG | Left Atrium | 2 | 20 | 3D |
| Brain Tumor (Cheng et al., 2015) | MRI | CLS | Tumor Types | 3 | 3,064 | 2D |

## 3.3 Evaluation Settings

In our analysis, we focused mainly on the "out-of-the-box" performance of DINOv2, where we trained classification or segmentation heads while keeping the backbone frozen. This resulted in preferable lightweight training that requires fewer labeled instances, computational resources, and training time. Additionally, we also performed end-to-end fine-tuning and parameter-efficient fine-tuning evaluations for performance comparison to this lightweight training paradigm. We describe our hyper-parameter settings in Appendix B, and we present all training logs in Section 7.

We experimented with the original DINOv2 ViT-g/14 and the three smaller distilled versions (ViT-L/14, ViT-B/14, and ViT-S/14). We compared the performance and generalizability of DINOv2 to supervised, self-supervised, and weakly-supervised models. Table 2 detailed all the models used.

Table 2: Models used. Description of the backbones used along with their parameter count and pre-training settings. "Used For" describes what we used each model for, and "# Images" describes the number of images in the pre-training dataset.

| Pretraining Method | Architecture | Dataset | Used For | # Images | # Params. | Citation |
|---|---|---|---|---|---|---|
| | | | Supervised | | | |
| | DenseNet201 | ImageNet1k | CLS | 1.3M | 20M | (Huang et al., 2018) |
| CLS | ResNet152 | ImageNet1k | CLS | 1.3M | 60M | (He et al., 2016) |
| | VGG19 | ImageNet1k | CLS | 1.3M | 144M | (Simonyan & Zisserman, 2015) |
| | ViT-L/16 | ImageNet21k | CLS, SEG | 14M | 300M | (Dosovitskiy et al., 2021) |
| SAM | ViT-L/16 | SA-1B | CLS | 11M | 300M | (Kirillov et al., 2023) |
| | | | Weakly-Supervised | | | |
| CLIP | ViT-L/14 | WIT-400M | CLS, SEG | 400M | 300M | (Radford et al., 2021) |
| OpenCLIP | ViT-H/14 | LAION-2B | CLS, SEG | 2,000M | 632M | (Ilharco et al., 2021) |
| | | | Self-Supervised | | | |
| MAE | ViT-L/16 | ImageNet1k | CLS, SEG | 1.3M | 300M | (He et al., 2021) |
| MSN | ViT-L/16 | ImageNet1k | CLS, SEG | 1.3M | 300M | (Assran et al., 2022) |
| | ViT-S/14 | LVD-142M | CLS | 142M | 21M | (Oquab et al., 2023) |
| DINOv2 | ViT-B/14 | LVD-142M | CLS | 142M | 86M | (Oquab et al., 2023) |
| | ViT-L/14 | LVD-142M | CLS, SEG | 142M | 300M | (Oquab et al., 2023) |
| | ViT-g/14 | LVD-142M | CLS, SEG | 142M | 1,100M | (Oquab et al., 2023) |

A critical part of our work is that we standardized the training and hyperparameter tuning pipeline for each model and evaluation task, with the goal of isolating model performance. In this way, we can gain a clearer understanding of which models perform best under the same evaluation settings.

**Disease Classification.** We performed four main types of experiments: kNN, linear-probing, few-shot learning, and fine-tuning. (1) kNN was performed on the normalized features of the last backbone layer. (2) For linear-probing, a single linear layer was attached on top of the backbone. (3) In few-shot learning, we trained a linear layer on top of frozen features. (4) For fine-tuning, we used both parameter-efficient fine-tuning methods like LoRA (Hu et al., 2021) and BitFit (Zaken et al., 2022) and end-to-end fine-tuning.

When performing linear-probing with ViT architectures (Dosovitskiy et al., 2021), the linear layer takes either the CLS token or the CLS token concatenated with the average of all patch tokens, depending on which method yielded higher performance in the validation set. In multi-labeled classification, we predicted each class as a binary and averaged the model's result across all classes, and we used the method described by (Zhang & Zhou, 2007) for kNN. For 3D volumes, the embeddings for all slices were averaged before being passed into the classification head.

**Organ Segmentation.** For organ segmentation evaluations, we kept the encoder frozen and attached a U-Net, hierarchical decoder on top. The U-Net decoder is made up of four blocks, where each block consists of one convolutional layer along with ReLU activation function and batch normalization. Skip connections were obtained from the previous four blocks of the transformer model and concatenated to the features at each U-Net layer, as is done in the classical U-Net architecture (Ronneberger et al., 2015).

We also compared the performance of DINOv2 to segmentation architectures that are commonly used in medical image analysis including U-Net and TransUnet, trained end-to-end in a supervised fashion, and experimented with using a lightweight single linear layer decoder as a comparison with the U-Net decoder. For processing 3D volumes, we segmented each slice independently.

**Comparison with medical-image pre-trained models.** It is important to note that we do not compare DINOv2 to domain specific models like MedCLIP (Wang et al., 2022), BiomedCLIP Zhang et al. (2024), MedSAM (Ma et al., 2023), or other medical image pre-trained models. The goal of this work is to evaluate whether DINOv2 pre-trained models can learn general-purpose representations in the medical domain that are useful across medical image analysis tasks like disease classification and organ segmentation, and across different radiology modalities like X-ray, CT, and MRI. All models used are pre-trained on natural images.

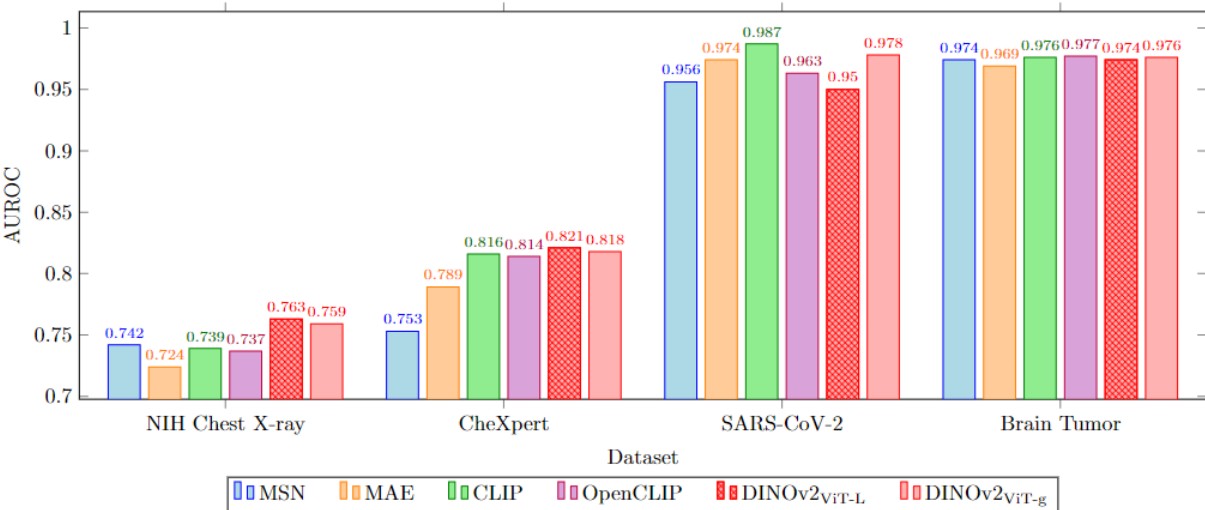

Figure 3: Linear probing for disease classification compared to self-supervised and weakly-supervised methods. The figure shows the performance of linear probing DINOv2 compared to other self-supervised and weakly-supervised models.

## 4 Results

In this section, we report our results across the different evaluation settings, tasks, and radiological modalities. We compare the performance of DINOv2 to weakly-supervised models including CLIP (Radford et al., 2021) and OpenCLIP (Ilharco et al., 2021), and self-supervised models including MAE (He et al., 2021) and MSN (Assran et al., 2022), and supervised models. We used the area under the operating receiver curve (AUROC) as a performance metric for classification tasks, and the average of the dice and jaccard scores as a metric for segmentation. We report only the best checkpoint across the epochs tested for each model.

In Section 4.1 and we will evaluate DINOv2 on disease classification and organ segmentation tasks. For segmentation, we will also show a comparison between using a linear layer decoder and a U-Net, hierarchical decoder on top of the frozen DINOv2 ViT-L/14 features. After that, in Section 4.2, we will analyze the cross-task generalizability of DINOv2 compared to the other models on disease classification and organ segmentation tasks. Then, in Section 4.3, we will explore the few-shot learning capability of DINOv2 on both disease classification and organ segmentation tasks. In Section 4.4, we perform parameter-efficient fine-tuning to compare the performance and efficiency of using PEFT with end-to-end fine-tuning and linear probing. Finally, 4.5 shows qualitative results of DINOv2 features on X-ray, CT, MRI modalities, and organ segmentation results of linear and U-Net decoders trained on top of frozen DINOv2 ViT-L/14 features.

### 4.1 Disease Classification and Organ Segmentation

We will start by analyzing the performance of DINOv2 ViT-g/14 and ViT-L/14 on linear-probing. Figure 3 shows the AUROC scores of both DINOv2 models compared to weakly-supervised and self-supervised models on four disease classification benchmarks. DINOv2 outperforms all other models on the more difficult large X-ray datasets and achieves performance closer to weakly-supervised models that are trained on much larger datasets. The ViT-L version of the model seemingly underperforms on 3D CT classification but is competitive with the ViT-g/14 version on all other classification tasks. Additionally, Table 3 shows the linear-probing performance of all DINOv2 models compared to supervised learning methods on X-ray, CT, and MRI disease classification datasets. DINOv2 performs on par or slightly better compared to supervised methods, and outperforms by a relatively larger margin commonly-used CNN supervised models.

Table 4 shows the kNN, linear-probing, and end-to-end fine-tuning results of DINOv2 compared to other supervised, weakly-supervised, and self-supervised methods on the NIH Chest X-ray and CheXpert datasets.

Table 3: Linear probing for disease classification compared to supervised methods. The Table compares supervised models with DINOv2 pre-trained models in Linear-probing settings on X-ray, CT, and MRI datasets. All models were tested under the same settings.

| Method | Architecture | NIH Chest X-ray | CheXpert | SARS-CoV-2 | Brain Tumor |
|---|---|---|---|---|---|
| | DenseNet201 | 0.735 | 0.795 | 0.973 | 0.960 |
| Supervised | ResNet152 | 0.718 | 0.779 | 0.936 | 0.948 |
| | VGG19 | 0.696 | 0.750 | 0.891 | 0.933 |
| | ViT-L/16 | 0.751 | **0.829** | **0.983** | 0.975 |
| | ViT-S/14 | 0.747 | 0.805 | 0.943 | 0.962 |
| DINOv2 | ViT-B/14 | 0.755 | 0.812 | 0.922 | 0.972 |
| | ViT-L/14 | **0.763** | 0.821 | 0.950 | 0.974 |
| | ViT-g/14 | 0.759 | 0.818 | 0.978 | **0.976** |

Table 4: DINOv2 performance comparison on large X-ray datasets. DINOv2 outperforms other methods on Linear-probing and fine-tuning but under performs on kNN evaluations. Multiple learning rates for tuning each backbone were tested, and only the best is shown.

| Method | Architecture | NIH Chest X-ray | | | CheXpert | | |
|---|---|---|---|---|---|---|---|
| | | kNN | Linear-probing | Fine-tuning | kNN | Linear-probing | Fine-tuning |
| | DenseNet201 | **0.675** | 0.735 | 0.769 | 0.783 | 0.795 | **0.882** |
| Supervised | Resnet152 | 0.668 | 0.718 | 0.752 | 0.766 | 0.779 | 0.868 |
| | VGG19 | 0.644 | 0.696 | 0.711 | 0.728 | 0.750 | 0.870 |
| | ViT-L/16 | 0.663 | 0.751 | 0.761 | 0.777 | **0.829** | 0.873 |
| MAE | ViT-L/16 | 0.659 | 0.724 | 0.743 | 0.785 | 0.789 | 0.821 |
| MSN | ViT-L/16 | 0.692 | 0.742 | 0.707 | **0.807** | 0.753 | 0.802 |
| CLIP | ViT-L/14 | 0.655 | 0.739 | 0.697 | 0.742 | 0.816 | 0.842 |
| OpenCLIP | ViT-H/14 | 0.659 | 0.737 | **0.770** | 0.744 | 0.814 | 0.847 |
| DINOv2 | ViT-L/14 | 0.663 | **0.763** | 0.717 | 0.771 | 0.821 | 0.786 |
| | ViT-g/14 | 0.659 | 0.759 | 0.769 | 0.768 | 0.818 | 0.848 |

Just like in linear probing, DINOv2 outperforms self-supervised methods in end-to-end fine-tuning but is competitive with weakly-supervised and supervised methods. Also, important to highlight that DINOv2 under-performs on kNN evaluations compared to other methods, even though its features were designed to maximize kNN results. This might be explained by the domain shift between natural images in the pre-training and medical images, making the out-of-the-box kNN evaluations more random.

Table 5: DINOv2 on organ segmentation. A comparison between using a frozen DINOv2 backbone and other commonly-used segmentation models initialized from scratch.

| Method | Architecture | MC | AMOS | MSD Heart | MSD Hipp | MSD Spleen |
|---|---|---|---|---|---|---|
| MAE | ViT-L/16 | 0.969 | 0.547 | 0.864 | 0.799 | 0.853 |
| MSN | ViT-L/16 | 0.961 | 0.407 | 0.810 | 0.788 | 0.784 |
| CLIP | ViT-L/14 | 0.955 | 0.445 | 0.762 | 0.748 | 0.747 |
| OpenCLIP | ViT-H/14 | 0.962 | 0.501 | 0.793 | 0.779 | 0.803 |
| | ViT-L/14$_{224}$ | 0.966 | 0.512 | 0.792 | **0.812** | 0.813 |
| DINOv2 | ViT-L/14$_{448}$ | **0.974** | 0.592 | 0.869 | 0.789 | 0.898 |
| | ViT-g/14$_{224}$ | 0.966 | 0.511 | 0.804 | 0.761 | 0.802 |
| | ViT-g/14$_{448}$ | 0.973 | **0.642** | **0.875** | 0.729 | **0.900** |

Moreover, we provide a comparison for segmentation tasks in Table 5. When comparing just image sizes of 224x224, DINOv2 performs the second-best, just behind MAE, which outperforms DINOv2 on all but one segmentation task. This can be explained given MAE's inherently pixel-level learning objective. An important point to note here is that all models except DINOv2 were pre-trained on image sizes of 224x224. At the end of the DINOv2 pre-training, it was adapted to image size 518x518, leading to worse performance on smaller image sizes due to positional encoding interpolation (Oquab et al., 2023). Even then, DINOv2 still outperforms weakly-supervised models and MSN pre-training. When using image sizes closer to DINOv2's pre-training, it outperforms all other methods. However, it is difficult to determine how much of this increased performance is due to reduced interpolation of positional encoding or to higher details resulting from the larger images.

Table 6: DINOv2 compared to segmentation architectures. A comparison between using a frozen DINOv2 backbone and other commonly-used segmentation models initialized from scratch. The parameter count given is when there is one output class.

| Method | Architecture | Trainable Params. | Total Params. | MC | AMOS | MSD Heart | MSD Hipp | MSD Spleen |
|---|---|---|---|---|---|---|---|---|
| Scratch | U-Net | 31M (100%) | 31M | 0.973 | 0.432 | **0.911** | 0.593 | 0.826 |
| | TransUnet | 324M (100%) | 324M | **0.974** | 0.535 | 0.892 | **0.821** | 0.855 |
| DINOv2 | ViT-L/14 | 17M (5%) | 317M | **0.974** | 0.592 | 0.869 | 0.789 | 0.898 |
| | ViT-g/14 | 38M (3%) | 1,138M | 0.973 | **0.642** | 0.875 | 0.729 | **0.900** |

**DINOv2 compared to segmentation architectures.** In Table 6 we compare U-Net decoders on top of frozen DINOv2 ViT-L/14 and ViT-g/14 features with U-Net and TransUnet models trained end-to-end from scratch. The results of all models are similar on the easier MC dataset, but DINOv2 outperforms the other U-Net and TransUnet on the more difficult AMOS multi-organ segmentation task even with a frozen encoder and less trainable parameters. On the MSD datasets, DINOv2 is competitive, but there is variability in the performance.

**U-Net vs linear decoders.** Table 4 shows a performance comparison between using a linear layer decoder and a U-Net decoder on top of frozen DINOv2 ViT-L/14 features on the organ segmentation datasets. The linear layer decoder evaluations are used to isolate the performance of the DINOv2 encoder, analogous to the purpose of kNN classification evaluations. On the MC dataset where the target mask is large, linear and U-Net performance is comparable, highlighting the strong out-of-the-box features of DINOv2. The gap in performance increases, however, on MSD datasets where target masks are usually much smaller, making them harder to predict with a single layer. Figure 6 shows qualitative segmentation for both methods.

## 4.2 Cross-task Generalizability Analysis

In this section, we evaluate whether DINOv2 can produce representations that are more generalizable across tasks, compared to other models. To accomplish this task, we plot in Figure 1 the relative performance of weakly-supervised and self-supervised methods across disease classification and organ segmentation. The horizontal axis shows the relative performance or ranking of each of the models on segmentation tasks, while the vertical axis does the same for classification tasks. The ranking is calculated as the average ranking across all segmentation or classification tasks, where rank 1 means the model performs the best relative to the other models. Specifically, we assigned a score between 1 and 6 for each model on each dataset, where a score of 6 means that the model achieved the highest result and 1 means the lowest. We averaged the scores for all datasets and plotted the ranking for each model based on this score, by assigning rank 1 to the highest score. An interesting observation is that models pre-trained with weakly-supervised learning perform well only on classification tasks, while MAE performs well only on segmentation. DINOv2 can generalize across both tasks and outperforms all other models for classification.

**DINOv2 cross-task generalizability compared to supervised methods.** Additionally, we carry out two experiments to compare the task-generalizability of DINOv2 with supervised methods. First, we compare the segmentation performance of a DINOv2 pre-trained ViT-L/14 with a ViT-L/16 pre-trained with supervised learning on ImageNet21k. Table 7 shows the results. DINOv2 outperforms the supervised ViT

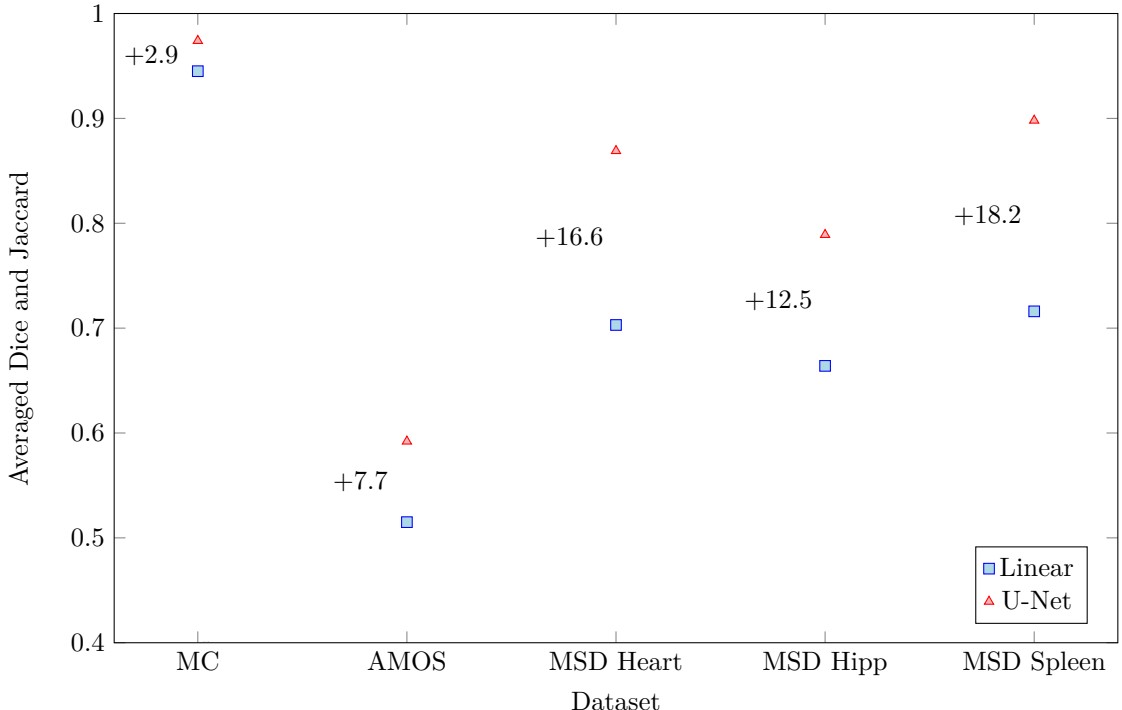

Figure 4: Linear vs. U-Net decoder. Comparison of the Linear and U-Net decoders on the four segmentation tasks used in this analysis. The backbones used in both is a DINOv2 ViT-L/14.

on 4 out of the 5 organ segmentation tasks, especially in the challenging AMOS multi-organ segmentation task.

Our second experiment compares the performance of the SAM image encoder with DINOv2 on classification tasks. SAM was trained for prompt-based segmentation and does not have a CLS token. To perform classification with SAM we averaged all the patch embeddings and treated the result as a CLS token. Table 8 shows the results. Images are of size 1024x1024 and only a subset of each dataset was used because of computational limits. DINOv2 significantly outperforms SAM on both datasets, highlighting the cross-task generalizability of the model.

Table 7: DINOv2 vs. ImageNet21k pre-trained ViT on organ segmentation. DINOv2 outperforms the supervised pre-trained ViT on 4 of the 5 tasks.

| Method | Architecture | Image Size | Montgomery County | AMOS | MSD Heart | MSD Hipp | MSD Spleen |
|---|---|---|---|---|---|---|---|
| Supervised | ViT-L/16 | 224 | 0.963 | 0.433 | 0.825 | 0.750 | 0.773 |
| DINOv2 | ViT-L/14 | 224 | 0.966 | 0.512 | 0.792 | 0.812 | 0.813 |
| | ViT-L/14 | 448 | 0.974 | 0.592 | 0.869 | 0.789 | 0.898 |

Table 8: SAM vs DINOv2 on X-ray on disease classification. Only a subset of the entire dataset was used, and in that subset all images were resized to 1024x1024 to fit into SAM's image encoder. The average of patch embeddings was used for classification for SAM.

| Method | Architecture | NIH Chest X-ray | CheXpert |
|---|---|---|---|
| SAM | ViT-L/16 | 0.714 | 0.792 |
| DINOv2 | ViT-L/14 | **0.755** | **0.816** |

### 4.3 Few-shot Learning

To measure DINOv2's ability to adapt to new distributions using a few labeled instances, we perform few-shot learning for both disease classification and organ segmentation on X-ray datasets.

At the top row of Figure 5, we start by comparing DINOv2 ViT-L/14 to weakly-supervised and self-supervised methods. The top-left subplot shows the performance on the NIH Chest X-ray disease classification dataset, while the top-right subplot shows the performance on the MC lung segmentation datasets. For disease classification, there is no clear trend when the number of patients used for each class is between 1 and 4, but when 8 patients are used, DINOv2 outperforms all other methods. For organ segmentation, DINOv2 outperforms all other methods from the start and is only worse than MAE when the entire dataset is used.

A similar trend can be observed at the bottom row of Figure 5, where we compare DINOv2 with supervised methods. The bottom-left subplot shows that DINOv2 outperforms other methods when the number of patients are 8 or more, but to a lesser degree compared to self-supervised and weakly-supervised methods. For organ segmentation, DINOv2 outperforms other models by a large margin when using less than eight instances, which is somewhat expected given it was pre-trained while the other segmentation models were not.

### 4.4 Parameter-efficient Fine-tuning

We experiment with parameter-efficient fine-tuning (PEFT) techniques on DINOv2 ViT-g/14, which, as a whole, contains 1.1 billion parameters. PEFT methods are used to enable efficient adaptation of large models to downstream tasks, usually achieving performance that is on par with end-to-end fine-tuning while requiring a lot less compute and memory. Previous work by Dutt et al. (2023) has highlighted the opportunity of employing PEFT to tune large foundation models for medical image analysis.

We employ two different PEFT techniques: LoRA (Hu et al., 2021) and BitFit (Zaken et al., 2022). LoRA is an additive method that inserts trainable decomposition matrices in the layers of a transformer, while BitFit is a selective method that unfreezes only the bias terms of the model. Table 9 shows a result and efficiency comparison between the two PEFT methods with a comparison to end-to-end fine-tuning and linear-probing on the NIH Chest X-ray and CheXpert datasets using the DINOv2 ViT-g/14 model.

Table 9: PEFT on DINOv2 ViT-g/14. Both LoRA and BitFit achieve results that are better than linear-probing while adapting less than 1% of the total parameters.

| Method | Trainable Params. (%) | NIH Chest X-ray | CheXpert |
|---|---|---|---|
| Fine-tuning | 1,100M (100%) | 0.769 | 0.848 |
| Linear-probing | 1,500 (1e-6%) | 0.759 | 0.818 |
| LoRA | 8M (0.7%) | 0.767 | 0.823 |
| BitFit | 0.8M (0.07%) | 0.768 | 0.817 |

### 4.5 Qualitative Results

In this section we will show qualitative results of DINOv2 features using principal component analysis (PCA) performed on DINOv2 patch features on X-ray, CT, and MRI scans, following the method delineated in (Oquab et al., 2023). We will also provide organ segmentation results of linear compared U-Net decoders.

**PCA visualization.** Figure 2 shows the first three PCA components. The PCA is computed between patches of images that are in the same column, and the first 3 components are shown for X-ray, CT, and MRI scans. Thresholding is used on the first PCA component to remove the background. Just like in natural images (Oquab et al., 2023), the colors of the three PCA components correspond well with the same parts of images in the same category. This is an easier task, however, compared to natural images, because there is less variability between examinations on medical images compared to natural images.

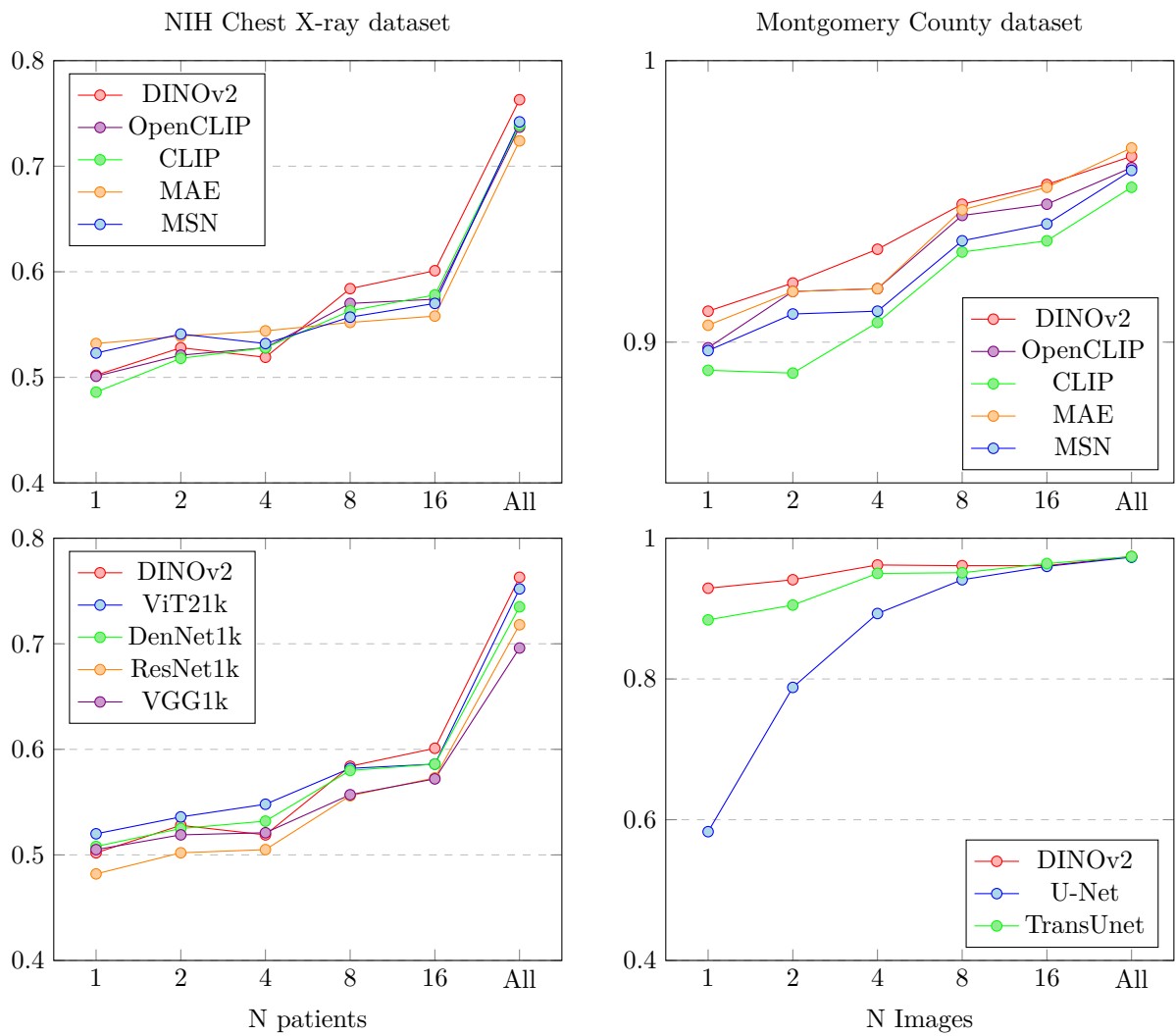

Figure 5: Few-shot disease classification and organ segmentation. The top row compares DINOv2 ViT-L/14 with weakly-supervised and self-supervised methods on the NIH Chest X-ray and MC datasets. The bottom row provides a comparison with supervised methods. For disease classification, there is no clear trend when the number of patients used for each class is between 1 and 4, but when 8 patients are used, DINOv2 clearly outperforms all other methods. For organ segmentation, DINOv2 outperforms all other methods from the start.

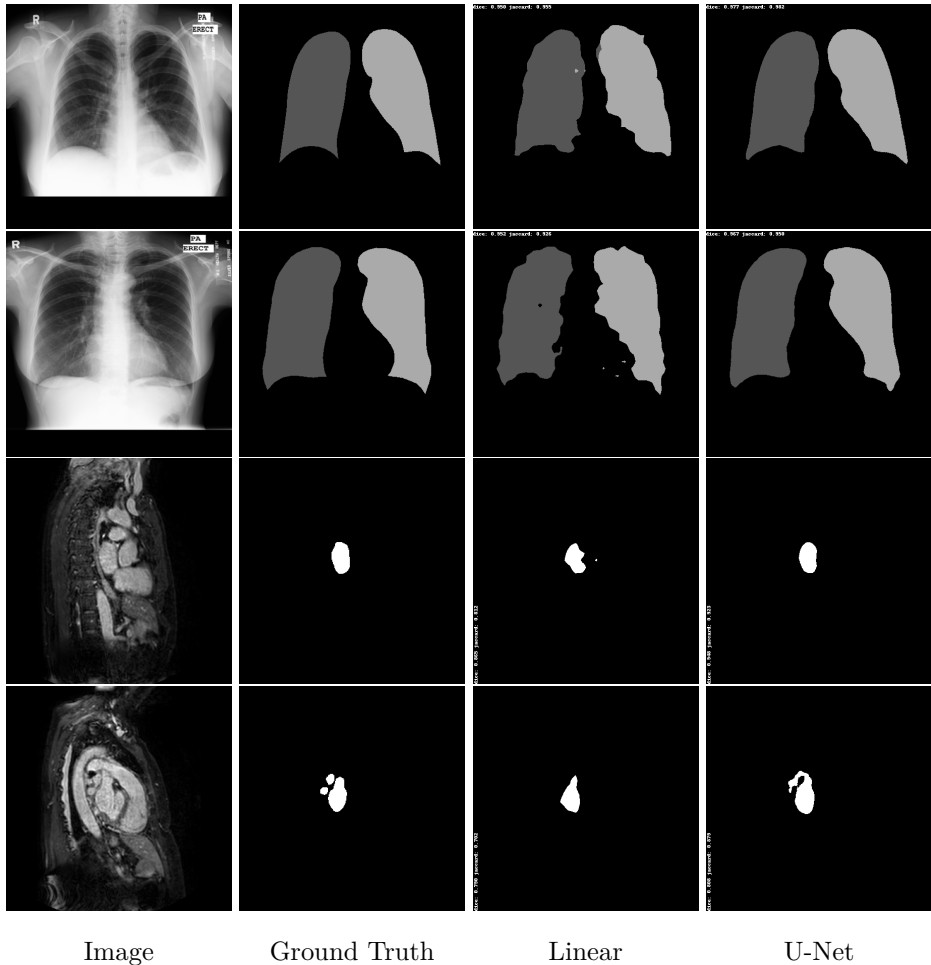

|       Image       |    Ground Truth    |      Linear      |      U-Net      |

Figure 6: Linear vs. U-Net visualization. The figure shows a qualitative comparison between segmentation masks generated by the linear layer and the U-Net decoder.

**U-Net and linear decoder visualization.** We also show a visualization of linear and U-Net decoders trained on top of DINOv2 ViT-L/14 features. The linear layer decoder performs surprisingly well, but is limited, especially on smaller masks, due to the smaller decoding map (32x32 pixels interpolated to 448x448) and less adjustable parameters. As expected, the U-Net segmentation results are smoother and represents the ground truth mask more accurately, but is still limited due to the frozen encoder.

## 5 Discussion and Future Direction

Foundation models have shown promise for reducing the data annotation problem and increasing model generalizability and robustness. Thus, they are a direction towards increasing performance and adoption of deep learning systems in healthcare. However, training a foundation model from scratch is extremely challenging for most institutions, demanding substantial amounts of well-organized data and computational resources. This challenge is particularly pronounced in the realm of medical image analysis, where data annotation is markedly more expensive than in other fields, and the data itself comprises high-dimensional volumes. This paper examines DINOv2, a foundation model trained on natural images, for medical applications. The DINOv2 pre-training approach is specifically promising given its ability to learn general-purpose representations and perform well out-of-the-box, without needing to fine-tune the encoder. We believe that using DINOv2 pre-training on medical data is a promising approach for future research aimed at building large-scale medical foundation models without supervision.

# 6  Conclusion

In this work, we examine DINOv2, a self-supervised foundation model pre-trained on 142 million natural images, for applications to radiology image analysis across X-ray, CT, and MRI modalities. We conclude that DINOv2 is a strong feature extractor across both disease classification and organ segmentation tasks, and outperforms traditional ImageNet pre-trained CNN methods and other weakly-supervised and self-supervised pre-trained ViTs.

# 7  Limitations and Ethical Concerns

This work compares DINOv2 to other models pre-trained on natural images. It doesn't however compare DINOv2 to models pre-trained on medical data, which could present interesting conclusions to the community, specifically on which open-source model is currently leading for cross-task generalizability.

We don't believe that there are privacy or other special ethical concerns in our work. All the datasets used are public and contain no patient identifiable information. Our released models are trained on a scale too small to be deployable or useful for any real world setting. We caution against using DINOv2 or our fine-tuned models for medical advice.

## Reproducibility

All the data used for this work is publicly available, and our train, validation, and test split will be made publicly available. All the training and validation logs, hyperparameters, and model weights will also be made publicly available.

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

# A  Data Preprocessing and Splitting

## A.1  Data Splitting

We split all our datasets into training, evaluation, and test sets. The evaluation set is used to select the optimal hyper-parameter, and is then combined with the training set for training our final model, which is then evaluated on the test set. For datasets that are already arranged in this way (like CheXpert), we used the provided sets. For datasets that only a public test or evaluation set (like NIH Chest X-ray), we used that set as our final test set and created our own evaluation set using data from the training set. The evaluation set was sampled using systematic sampling. Our ratio of train and evaluation sets vary greatly, depending on the size of the original dataset. We publicly provide all our splits in the Google Drive and GitHub.

## A.2  Preprocessing

We followed the standard preprocessing techniques for the different radiological modalities. For AMOS CT scans, we clipped the image pixel values to a minimum of -1024 and a maximum of 600. For MRI scans, we clipped the lowest and highest to the 5% and 95% percentile, respectively. We also used data-augmentation to boost performance and increase generalizability. For classification training, we used a random scaling in the range of [0.75, 1], and random horizontal flopping with $p = 0.5$. For segmentation, we used stronger augmentation. We used horizontal and vertical flipping with $p = 0.25$ and random rotation with maximum of 90 degrees.

# B  Training Settings

## B.1  Fine-tuning

For our fine-tuning evaluations, we used a batch size of 128 for the smaller CNN models and a batch size of 32 for the larger ViTs. We also used two different learning rates, one for the newly initialized classifier layer and one for the backbone. For the classifier learning rate, we selected the learning rate that achieved the best performance in the linear-probing evaluations. For the backbone, we spanned the following: $\{5e^{-4}, 1e^{-4}, 5e^{-5}, 1e^{-5}, 5e^{-6}, 1e^{-6}, 5e^{-7}, 1e^{-7}\}$. We noticed that lower learning rates usually worked better for larger DINOv2 models. We used no average pooling for the patch tokens, and used the last layer's CLS token to make predictions. We used 20 training epochs for the NIH Chest X-ray dataset and 10 epochs for CheXpert.

## B.2  Linear-probing and kNN

We mainly followed the original DINOv2 settings for our linear-probing evaluations. Namely, we initialized multiple linear layers that are simultaneously trained using different hyperparameters. This means that the backbone output, which is the most expense to generate, is being passed to multiple independent linear layers that are trained together, making the hyperparameter tuning process more efficient. For the actual hyperparameters we tested, we selected learning rates from $\{1e^{-1}, 1e^{-2}, 5e^{-2}, 1e^{-3}, 5e^{-3}, 1e^{-4}, 5e^{-4}, 1e^{-5}\}$. For each learning rate, we also tested using the output from 1 or 4 of the last ViT layers. Moreover, for each setting, we also tested using average pooling of the patch tokens. We used a batch size of 128. Our final parameters selected for testing are the ones that perform the best on the validation. We used different evaluation and training epochs for each dataset, which we make available in our shared Google Drive.

for kNN evaluations, we presented results for the best value out of $k = \{100, 200, 400, 800, 1000\}$. For multi-label kNN, we used the approach in (Zhang & Zhou, 2007).

## B.3  PEFT

For LoRA evaluations, we used a learning rate for $2e^{-4}$ for tuning Lora-introduced parameters, and a learning rate of $5e^{-3}$ for the linear classifier. We also used a LoRA rank $r = 48$ and $\alpha = 16$, with dropout $p = 0.1$. We used the Hugging Face PEFT library for implementation (Mangrulkar et al., 2022).

## B.4    Segmentation

For all segmentation evaluations, we optimized the dice loss because we observed that it is more stable than the combined dice and cross-entropy loss. We scanned the learning rates $\{1e^{-2}, 5e^{-2}, 1e^{-3}, 5e^{-3}\}$. We used a batch size of 16. We used different evaluation and training epochs for each dataset, which we make available in our shared Google Drive.

