# OpenReview forum: "Evaluating General Purpose Vision Foundation Models for Medical Image Analysis: An Experimental Study of DINOv2 on Radiology Benchmarks"
_TMLR — Rejected by TMLR_

### Review · Reviewer_WCee · 2024-10-06

**Summary Of Contributions:**

This paper evaluates the performance of DINOv2 for radiology image analysis tasks comprehensively. The evaluation spans across various tasks and settings. They find that DINOv2 is better than weakly-supervised and self-supervised methods, and it is competitive with supervised methods in organsegmentation tasks and disease classfication. In addition, this paper discusses the cross-task generalizability of DINOv2.

**Audience:**

No

**Claims And Evidence:**

No

**Requested Changes:**

I have two suggestions:
1. It would be helpful to provide more justification for the current setting, tasks, and model choice.
2. In addition to 1, it is more important to make the scope of the paper more general.For example, benchmarking more models is a good way.

**Strengths And Weaknesses:**

Paper Strength:
- This paper is self-contained and provides a comprehensive evaluation on various radiology benchmarks.They offer insights on understanding the performance of DINOv2 in different scenarios.
- They discusses the cross-task generalizability and highlights its strengths for DINOv2.

Paper Weaknesses:

(1) Actually, I don’t understand why this setting was chosen. Although the author spent a lot of space explaining their motivation on page 3, I still can’t be convinced by the author’s explanation. For example, I have a few questions:
- Why must it be the DINOv2 model? Why can’t it be USM or SAM and any other models? The paper mentioned that such models can reduce annotated data requirements, but if only one model is studied, whether the conclusion can be generalized to other models is a problem.
- What confuses me as well is that the author mentioned in section 1.2 that the training set of DINOv2 is LVD-142M, which has almost no medical images, so I don’t understand why the DINOv2 model is studied. If we start from a practical point of view, shouldn't we use a foundation model trained in the medical field, such as MedSAM?
- I appreciate the author's contribution, but I am confused about the motivation of the paper, which does not seem to be well-motivated.

(2) Section 1 and Section 2 seem to be dedicated to clarifying the differences between the three foundation models, but I don't think this is important because the focus of this paper is radiology (not general) benchmarks.

---

> ### Author Response · Authors · 2024-11-07
>
> We thank the reviewer for their thoughtful comments.
>
> **Why must it be the DINOv2 model? Why can’t it be USM or SAM and any other models? The paper mentioned that such models can reduce annotated data requirements, but if only one model is studied, whether the conclusion can be generalized to other models is a problem.**
>
> The reason we chose DINOv2 is that the model is advertised as being a state-of-the-art general purpose feature extractor that can work across a variety of tasks and domains [1]. At the time of the study, it was the *best* open-source model, outperforming other models trained on 10x more data. Other models like SAM (segmentation) or USM (speech) are *designed* only for their specific tasks. In our paper, we included evaluations with other general-purpose models like MAE, OpenCLIP, etc. This work shows that, because DINOv2 is a general purpose model, it can be used even for medical domain as a feature extractor, motivating further research that relies on the model for other tasks like medical image registration [2].
>
> **What confuses me as well is that the author mentioned in section 1.2 that the training set of DINOv2 is LVD-142M, which has almost no medical images, so I don’t understand why the DINOv2 model is studied. If we start from a practical point of view, shouldn't we use a foundation model trained in the medical field, such as MedSAM?**
>
> Similar to our above response, DINOv2 is advertised as being a general purpose model, suitable across different domains. This motivated its usage in medical image registration [2], 3D medical image classification [3], and even in Meta's brain decoding model [4]. All the previous papers use DINOv2 out-of-the-box, without being trained on medical images. While other models like MedSAM are trained on medical images, they are specifically trained for segmentation, not general purpose feature extraction, which makes them ill-suited for our study. However, the capabilities of MedSAM for feature extraction is still an interesting question for future study.
>
> **Section 1 and Section 2 seem to be dedicated to clarifying the differences between the three foundation models, but I don't think this is important because the focus of this paper is radiology (not general) benchmarks.**
>
> We believe that Section 1.1 is directly related to the paper's motivation, describing why DINOv2 is suitable for general-purpose feature extraction, and the difference between it and task-specific foundation models like SAM. Moreover, Section 2 contains important background information about what makes DINOv2 special compared to other foundation models, and references many related works that use SSL or weakly-supervised learning for radiology, and what makes them less suitable for a study like ours compared to DINOv2 (model and data scale).
>
> **it is more important to make the scope of the paper more general.For example, benchmarking more models is a good way.**
>
> Even though the paper mainly focuses on DINOv2, we experimented with more than different 13 models, as is shown in our Table 2. These experiments are computationally expensive to run especially for larger models like the ViT-L and ViT-g, and especially since we first run validation experiments to find the optimal hyperparameters before running our final test run, ensuring a fair comparison. We believe that, currently, the paper balances having a general scope of benchmarking multiple models as well as being specific enough to answer a direct question of "How good is DINOv2 for medical images?"
>
> We would be happy to know what the claims in the paper that are not supported by evidence, so that we can directly adress them. Furthermore, we also think this paper could be of interest to the TMLR audience since it benchmarks currently leading open-source models across a variety of medical image analysis tasks, providing valuable insights on which SSL approach could be further utilized to pre-train a large image encoder on medical datasets.
>
> References:
>
> [1] Oquab, M., Darcet, T., Moutakanni, T., Vo, H., Szafraniec, M., Khalidov, V., Fernandez, P., Haziza, D., Massa, F., El-Nouby, A. and Assran, M., 2023. Dinov2: Learning robust visual features without supervision. arXiv preprint arXiv:2304.07193.
>
> [2] Song, X., Xu, X. and Yan, P., 2024, October. DINO-Reg: General Purpose Image Encoder for Training-Free Multi-modal Deformable Medical Image Registration. In International Conference on Medical Image Computing and Computer-Assisted Intervention (pp. 608-617). Cham: Springer Nature Switzerland.
>
> [3] Kiechle, J., Lang, D.M., Fischer, S.M., Felsner, L., Peeken, J.C. and Schnabel, J.A., 2024. Graph Neural Networks: A suitable Alternative to MLPs in Latent 3D Medical Image Classification?. arXiv preprint arXiv:2407.17219.
>
> [4] Benchetrit, Y., Banville, H. and King, J.R., 2023. Brain decoding: toward real-time reconstruction of visual perception. arXiv preprint arXiv:2310.19812.

---

### Review · Reviewer_V58q · 2024-10-21

**Summary Of Contributions:**

This paper presents an experimental study of DINOv2 on radiology benchmark: including classification, segmentation tasks, few-shot, and PEFT learning strategy. The comprehensive results show the effectiveness of DINOv2 on medical images.

**Audience:**

Yes

**Claims And Evidence:**

No

**Requested Changes:**

* Comparing to foundation models in medical imaging to provide a more holistic understanding.
* More challenge benchmarks are needed to demonstrate the conclusion.

**Strengths And Weaknesses:**

## Strengths

* The paper is easy to follow and the experiment is comprehensive, covering classification, segmentation, and various learning strategies.

## Weakness:
1. The novelty is very limited. It conducts experiments with DINOv2 on different datasets, merely confirming an existing conclusion. Additionally, what are the advantages of DINOv2 compared to foundation models in medical imaging? I do not agree that it is appropriate to omit a comparison with foundation models in the medical domain.

2. The selected benchmark is not convincing. The results on SARS-CoCoV-2 and Brain Tumor datasets are nearly 100%, which suggests that these may not be suitable benchmarks to effectively demonstrate the model's performance.

---

> ### Author Response · Authors · 2024-11-07
>
> We thank the reviewer for taking time to review our paper.
>
> **The novelty is very limited. It conducts experiments with DINOv2 on different datasets, merely confirming an existing conclusion. Additionally, what are the advantages of DINOv2 compared to foundation models in medical imaging? I do not agree that it is appropriate to omit a comparison with foundation models in the medical domain.**
>
> The key point is that we comprehensively evaluate the model for *medical images.* We showed that DINOv2 can be used for medical image analysis tasks, which was not an existing conclusion. This result is seen as motivation for other works that use DINOv2 features in medical imaging for other tasks like medical image registration [1]. We also agree that evaluations with medical foundation models would've been helpful, but we don't believe that it takes away from this work. The goal of this work is to see if DINOv2 is a strong feature extractor for medical images compared to other commonly used models (ImageNet pre-trained models are still heavily used in the field [2,3]). Furthermore, as is discussed in the paper, there aren't many open-source foundation models for medical images trained at a similar scale as DINOv2, at the time of the study. Most models are either task specific, modality specific, or not open-source. Though again, we agree that comparisons with models pre-trained on medical images would've been insightful, even if they are not the main purpose of the paper. We are willing to add them if we are granted more time.
>
> **The selected benchmark is not convincing. The results on SARS-CoCoV-2 and Brain Tumor datasets are nearly 100%, which suggests that these may not be suitable benchmarks to effectively demonstrate the model's performance.**
>
> We completely agree with you, results on these benchmarks are not conclusive because they are easier tasks. However, if you look at our Table 1, you are going to notice that we try to include a dataset for classification and segmentation for each modality we have. The brain tumor dataset was the *only* acceptable public MRI classification dataset we found (others were either way too small, disorganized, etc.) . The same thing goes with SARS-CoV-2. We included these two datasets for completeness, and we don't base our conclusion merely on them--we show 7 others that are more difficult.
>
> References:
>
> [1] Song, X., Xu, X. and Yan, P., 2024, October. DINO-Reg: General Purpose Image Encoder for Training-Free Multi-modal Deformable Medical Image Registration. In International Conference on Medical Image Computing and Computer-Assisted Intervention (pp. 608-617). Cham: Springer Nature Switzerland.
>
> [2] Kim, H.E., Cosa-Linan, A., Santhanam, N. et al. Transfer learning for medical image classification: a literature review. BMC Med Imaging 22, 69 (2022). https://doi.org/10.1186/s12880-022-00793-7
>
> [3] Salehi AW, Khan S, Gupta G, Alabduallah BI, Almjally A, Alsolai H, Siddiqui T, Mellit A. A Study of CNN and Transfer Learning in Medical Imaging: Advantages, Challenges, Future Scope. Sustainability. 2023; 15(7):5930. https://doi.org/10.3390/su15075930

---

### Review · Reviewer_8a9Q · 2024-10-29

**Summary Of Contributions:**

This paper focuses on developing vision-only foundation models for the medical domain using DINOv2 for disease classification and organ segmentation. It conducts extensive evaluations of DINOv2 across various scenarios and radiology modalities, including X-ray, CT, and MRI, considering both low (few-shot) and high data settings. The performance of DINOv2 is benchmarked against other models like CLIP and SAM. For disease classification, evaluations include kNN, linear probing, few-shot learning, and different fine-tuning methods. For organ segmentation, the study compares lightweight and non-lightweight decoders while keeping the DINOv2 backbone frozen. The paper claims to be the first comprehensive analysis of DINOv2 on medical benchmarks for both tasks across different radiological examinations.

**Audience:**

Yes

**Broader Impact Concerns:**

None noted.

**Claims And Evidence:**

Yes

**Requested Changes:**

- The authors say that they average the patches and use it as a CLS token in SAM. Why not just discard the CLS token in DINOv2 instead? How does taking average of all the tokens improve performance of SAM instead of other methods, like taking the mode or max?
- Add some analyses following experimental results, explaining the observed trends in results.
- The authors could try adding a few more qualitative results comparing with other methods like MAE, CLIP.
- The motivation of the work should be clarified in a better way that justifies the use of DINOv2 over MAE or similar other methods pre-trained on smaller datasets.

**Strengths And Weaknesses:**

Strengths:

- The motivation for using DINOv2 seems appropriate in my opinion when not compared with other frameworks. However, the term "General Purpose Foundation Models" does not fit in that case, as the authors are exploring only one FM, that is, DINOv2. (Please look at "Requested Changes" for further comment on the motivation)
- This work covers different types of radiological datasets, like XRay, CT, MRI, and a lot of datasets as well, which is a good way to show the effectiveness of DINOv2 on radiological data.
- This work covers several different types of tasks including PEFT.


Weakness:
- The "Methods" column in the tables is confusing. The authors have not described which term stands for what and carries what significance in relevance to the task. For example, the first few rows say that the authors have used the method 'CLS' for 'CLS' task. Please clarify.
- Several experimental results lack a following analysis. Some sections are just tables of results with the interpretation being left to the readers.
- There are some errors in citations in the manuscript. _\citet_ used in place of _\citep_.
- Too few qualitative results.
- The motivation for using DINOv2 becomes gradually less significant when compared to methods like CLIP, OpenCLIP, MAE. If it is a matter of competitive performance, why not use MAE which is pre-trained on a much smaller dataset (ImageNet1K) than LVD-142M. Does not that make MAE more efficient, if it requires less pre-training data to achieve comparable performance with DINOv2 which is trained on an enormous dataset.

---

> ### Author Response · Authors · 2024-11-10
>
> We appreciate the review for their thoughtful comments.
>
> **The "Methods" column in the tables is confusing. The authors have not described which term stands for what and carries what significance in relevance to the task. For example, the first few rows say that the authors have used the method 'CLS' for 'CLS' task.
> Please clarify.**
>
> Fixed. We changed it to "Pretraining Method," which is much clearer now. So for the ones under CLS, it was pretrained using classification and then in the "Used for" column, it indicates that we are also using it for CLS here.
>
>
> **"Several experimental results lack a following analysis. Some sections are just tables of results with the interpretation being left to the readers."**
>
> We agree, and this was mainly because some of the tables/results are difficult to interpret/explain. However, the most important insight in the paper is presented in Figure 1, where we use the results from the multiple tables to show that some methods (MAE) work better on dense prediction tasks like segmentation, others (CLIP) work better on high-level tasks, while DINOv2 is able to generalize to both.
>
> **"There are some errors in citations in the manuscript. \citet used in place of \citep."**
>
> We used \citet when directly referencing other works, like in "Cohen et al. (2020) have attempted to train..." We used this format following the TMLR style file, "When the authors or the publication are
> included in the sentence, the citation should not be in parenthesis, using \citet{} (as in “See Hinton et al.
> (2006) for more information.”). Otherwise, the citation should be in parenthesis using \citep{} (as in “Deep
> learning shows promise to make progress towards AI (Bengio & LeCun, 2007).”)."
>
> **Too few qualitative results.**
>
> Because of the number of experiments we ran, we aimed to present general themes/trends (like in our Figures 1, 2, and 3) instead of specific results for some examples. This is why we opted to not show much qualitative results, though we would be willing to add them before acceptance.
>
> **The motivation for using DINOv2 becomes gradually less significant when compared to methods like CLIP, OpenCLIP, MAE. If it is a matter of competitive performance, why not use MAE which is pre-trained on a much smaller dataset (ImageNet1K) than LVD-142M. Does not that make MAE more efficient, if it requires less pre-training data to achieve comparable performance with DINOv2 which is trained on an enormous dataset.**
>
> The goal of the study is to benchmark DINOv2 against other self-supervised learning and foundation models. The motivation for us using DINOv2 for this study has been stated in the paper and in our response to the other comments--because the model is/was SoTA and was advertised as being a general purpose model that works across tasks/domains. We do not try to motivate using DINOv2 for other tasks in medical image analysis, we simply benchmark it and provide insights on its performance. One of our insights has been that DINOv2 can actually work across different tasks much better than MAE, CLIP, and OpenCLIP. When comparing *pre-training methods* however (MAE vs. DINOv2 for examples) our study offers no insights because it is obviously an unfair comparison when DINOv2 is pretrained on 100x more data. However, being trained on more data does not matter in this case since we already have the pretrained model, and we want to benchmark it--finetuning takes the same time for all models.
>
> **The authors say that they average the patches and use it as a CLS token in SAM. Why not just discard the CLS token in DINOv2 instead? How does taking average of all the tokens improve performance of SAM instead of other methods, like taking the mode or max?**
>
> ًWe could've added a CLS token for SAM or used averaged pooling for DINOv2. Previous research has shown that averaging embedding tokens actually performs similar to using a CLS token, so it shouldn't really matter [1]. I am not sure about using the max.
>
>
> References
>
> [1] Dosovitskiy, A., 2020. An image is worth 16x16 words: Transformers for image recognition at scale. arXiv preprint arXiv:2010.11929.

---

### Decision · Action_Editor_Zo9P · 2024-12-04

**Recommendation:** Reject

**Comment:**

**I. Paper Summary (by Reviewers)**

This paper focuses on developing vision-only foundation models for the medical domain using DINOv2 for disease classification and organ segmentation. It conducts extensive evaluations of DINOv2 across various scenarios and radiology modalities, including X-ray, CT, and MRI, considering both low (few-shot) and high-data settings. The performance of DINOv2 is benchmarked against other models like CLIP and SAM.


**II. Reviewer Feedback, Key Issues, and Discussions**

**All the reviewers unanimously rejected or weakly rejected the current version and raised some common concerns**. I would like to have a discussion on these issues.

Concern #1: The motivation for choosing DINOv2 in the context of medical imaging analysis
The major concern lies in the explicit rationale for considering DINOv2 in medical imaging. The reviewers did not simply raise concerns about DINOv2 itself but also about the comparison baselines like CLIP, OpenCLIP, and MAE (as presented in Fig. 1).
- Author’s Rebuttal: The goal of the study is to benchmark DINOv2 against other self-supervised learning and foundation models. The motivation for using DINOv2 for this study has been stated in the paper and in our response to other comments—because the model is/was SoTA and was advertised as a general-purpose model that works across tasks/domains. We do not try to justify using DINOv2 for other tasks in medical image analysis; we simply benchmark it and provide insights into its performance.
- AE’s Thoughts: I do not think benchmarking DINOv2 against other self-supervised learning and foundation models makes sufficient sense. This objective is quite awkward. As the reviewer concerns indicate, if the paper’s goal is to comprehensively evaluate the effectiveness of DINOv2’s performance in different medical imaging tasks, this can be viewed as a reproducibility paper on DINOv2. However, the additional comparisons with other SOTA methods make the contributions quite convoluted, such that it seems the main contribution is not reproducibility. Based on this, I agree that the paper’s scope is unclear.

Concern #2: The title is too broad. The title, “Evaluating General Purpose Vision Foundation Models for Medical Image Analysis: An Experimental Study of DINOv2 on Radiology Benchmarks,” is too general.

- AE’s Thoughts: I agree that this title is too broad, creating a gap between the actual experimental results and the claims in the title. This may be why some reviewers have concerns about using other foundation models. Therefore, I believe this concern is valid.


Concern #3: Limited revisions from the previous submission
- AE’s Thoughts. This is a resubmission. I further compared the current resubmission with the one from the last round. It seems like very minor revisions were made, whereas most prior issues remain largely unsolved.

**III. Decision**

Based on the current reviewers’ recommendations, my personal reading, and the reviews from the last round, I think we cannot accept this resubmission. The authors are strongly encouraged to incorporate feedback from both rounds of reviews to prepare a new submission.

**Audience:**

Yes

**Claims And Evidence:**

No, See the comments